ecology, palaeontology

extinction rates, origination rates, taxon longevity, resilience, capture-recapture modelling, survivorship

**Author for correspondence:**
Björn Kröger
e-mail: bjorn.kroger@helsinki.fi

# The evolutionary dynamics of the early Palaeozoic marine biodiversity accumulation

Björn Kröger[1], Franziska Franeck[2] and Christian M. Ø. Rasmussen[3,4]

[1]Finnish Museum of Natural History, University of Helsinki, Helsinki, Finland
[2]Natural History Museum, University of Oslo, Oslo, Norway
[3]Natural History Museum of Denmark, University of Copenhagen, Copenhagen, Denmark
[4]Center for Macroecology, Evolution and Climate, University of Copenhagen, Copenhagen, Denmark

BK, 0000-0002-2427-2364; FF, 0000-0002-7909-1800; CMØR, 0000-0003-2982-9931

The early Palaeozoic Era records the initial biodiversification of the Phanerozoic. The increase in biodiversity involved drastic changes in taxon longevity, and in rates of origination and extinction. Here, we calculate these variables in unprecedented temporal resolution. We find that highly volatile origination and extinction rates are associated with short genus longevities during the Cambrian Period. During the Ordovician and Silurian periods, evolutionary rates were less volatile and genera persisted for increasingly longer intervals. The 90%-genus life expectancy doubled from 5 Myr in the late Cambrian to more than 10 Myr in the Ordovician–Silurian periods. Intervals with widespread ecosystem disruption are associated with short genus longevities during the Cambrian and with exceptionally high longevities during the Ordovician and Silurian periods. The post-Cambrian increase in persistence of genera, therefore, indicates an elevated ability of the changing early Palaeozoic marine ecosystems to sustainably maintain existing genera. This is evidence of a new level of ecosystem resilience which evolved during the Ordovician Period.

## 1. Introduction

The spectacular early Palaeozoic rise in taxonomic richness of marine ecosystems continues to be a focus point of palaeobiological research [1–8]. It featured two distinct events of accelerated biodiversity accumulation, namely the Cambrian explosion (CE) and the Great Ordovician Biodiversification Event (GOBE). In addition, it contained a number of major crises during the Late Ordovician mass extinctions (LOME) [8].

A growing body of evidence suggests that the timing and intensity of the early Palaeozoic biodiversity accumulation was associated with changes in global temperature and oxygen levels [8–13]. However, the mechanisms linking, e.g. change in habitat space [12,14], spread of oxygen minimum zones [15,16], and extent of primary production [17,18] with biodiversity remain elusive [19].

Global biodiversity accumulation results from a combined process of origination and extinction of taxa, or viewed from a different perspective, it builds as a function of longevity of newly originating taxa. Hence, knowledge on taxon longevity and origination/extinction rates is essential to make inferences about the mechanisms of biodiversity accumulation. Many studies on evolutionary rates exist at the Phanerozoic and Palaeozoic scale and at the family and genus level (e.g. [4,20–28]). Longevity and survivorship rates have previously also been the focus of interest (e.g. [29–31]).

Rates of origination and extinction ultimately determine the probability of a taxon (here, a genus) to survive until a time *t* [29,30]. This relationship should not lead to the conclusion that analyses of longevity and evolutionary rates are redundant. Evolutionary rates inform about the volatility of the evolutionary change at a

*Proc. R. Soc. B* **286**: 20191634

given time interval, but they are agnostic about the specific composition of the rates of the individual genera and their life history. Identical evolutionary rates can be produced by originations and extinctions of long-living and short-living genera. Extinctions can preferably affect genera that persisted for a long time or, by contrast, genera that originated shortly before. Conversely, originations may result in long-lasting genera or short-living genera. The ecological mechanisms behind these different scenarios differ drastically and periods of ecosystem disturbance or resilience may remain unnoticed when only described by evolutionary rates.

Here, we present new estimates of rates of origination and extinction at the genus level with an unprecedented temporal resolution, based on a time binning established in Rasmussen *et al.* [8]. Additionally, we present for the first time per time bin estimates of longevity, taxon age, and taxon life expectancy of early Palaeozoic marine genera. Our results allow for a differentiation between taxonomic turnover and genus persistence, that again enables an evaluation of time-specific ecosystem resilience (i.e. the ability of a system to absorb changes and still persist, *sensu* Holling, [32]) as a factor of biodiversity accumulation.

## 2. Methods

We based our calculations on a sum of 173 293 genus-level Cambrian to Silurian fossil occurrences downloaded from the Paleobiology Database (PBDB, https://paleobiodb.org/#/, download 30 January 2019) and an additional download of 545 449 post-Silurian genus level occurrences from the PBDB (download 02 February 2019). The occurrences were binned into 53 Cambrian–Silurian time intervals with an average duration of 2.3 Myr following [8] and into post-Silurian stage intervals using the binning scheme of the PBDB (https://paleobiodb.org/data1.2/intervals/list.txt?scale=1, accessed 6 July 2019). Details of the data filtering and methodology of time binning and biodiversity calculations have been published in [8]. We estimated genus richness based on the capture-recapture model (CR) approaches [33,34] by fitting the Jolly–Seber model following the POPAN formulation [35]. We calculated relative diversification rates by dividing the richness difference between a time bin and its previous time bin with the richness of the respective time bin ($(n_{gen(t)} - n_{gen(t-1)})/n_{gen(t)}$). With $n_{gen}$ being the number of genera, $t$ being the time bin of interest, and $t - 1$ being the previous time bin.

Additionally, we estimated survival and seniority probabilities based on the CR-approach using the Pradel model [36], which were transformed into extinction and origination rates, following the transformation from probabilities into rates described in [37]. The method estimates survival, seniority, and sampling probabilities, which we turn into rates, to account for uneven sampling intervals (see electronic supplementary material, and [34] for details of the method). For comparison of our CR-modelling results with more conventional rate estimations, we calculated origination and extinction rates with the turnover rate metric of Alroy [38] as implemented in the R-package divDyn [39] (see electronic supplementary material).

We estimated genus age, genus life expectancy, and genus longevity indirectly by calculating forward and backward survivorships of cohorts of genera occurring in each time bin. The duration needed to reach the full diversity of genera occurring in each time bin is our measure of backward survivorship ($l_{bw}$) and can be read as a measure of genus age. The subsequent lifetime of the set of genera occurring in a time bin, is our measure of forward survivorship ($l_{fw}$) and can be read as a measure of life expectancy. Long-life expectancies of genera indicate a long persistence of the ecological relationships established among these genera. Hence, we interpreted $l_{fw}$ as an indicator of ecosystem

resilience (where resilience determines the persistence of relationships [32]). The backward survivorship can be read as a measure for the age structure of the genera of a time bin and reflects the history of the ecosystems. The sum of $l_{bw}$ and $l_{fw}$ is our overall longevity ($l_o$), which is a wrapper representing the past and the future of the genera that existed during each time bin.

Our longevity calculations are based on CR-modelled richness curves of the cohorts of genera occurring in each time bin of interest ($t_i$). In this calculation, a 100% richness always occurs in $t_i$ and the modelled richness always increases in time bins ($t_{i-n}$) preceding $t_i$ and decreases in the time bins ($t_{i+n}$) posterior to $t_i$. We determined the antecedent and posterior time bins containing 50%, 70%, and 90% $t_i$-richness levels and calculated $l_{bw}$ and $l_{fw}$ as the maximum time ranging from $t_i$ towards these time bins.

The complete algorithm and relevant results are recorded in R-code and can be downloaded at https://doi.org/10.5281/zenodo.3365505.

## 3. Results

### (a) Origination and extinction rates

Our estimated origination and extinction rates reveal basic differences between Cambrian and post-Cambrian evolutionary dynamics (figure 1c). The Cambrian rates are on average much higher than the post-Cambrian rates. Fluctuations of rates between time bins are much greater in the Cambrian Period. The generally decreasing Cambro–Ordovician rates trend was known already from curves with lower stratigraphic resolution [4,25–27]. Additionally, data from trilobites evidenced distinct differences in survivorships between Cambrian and Ordovician cohorts [30]. Our results show that this trilobite survivorship change reflects a more general pattern and that there is a strong change at the Cambro–Ordovician boundary. The significance of the trend change can partially also be demonstrated with a time series changepoint analysis, where a single changepoint of the origination rate time series occurs at the Cambro–Ordovician boundary (electronic supplementary material, figure S1).

Notably, the origination and extinction rates calculated with Alroy's [38] turnover rate metric show a less rapid but more continuous decrease at the Cambro–Ordovician boundary and continued to drop until the beginning of the Middle Ordovician (electronic supplementary material, figure S2), similar to, e.g. in Bambach *et al.* [26]. Bambach's [26] estimations and Alroy's [38] turnover metric result in instantaneous rates that do not account for the length of the time bins. The high estimates in the earliest Ordovician time bins in Alroy's and Bambach's calculations are therefore probably an effect of the poorly constrained timing of these intervals. The relatively long Early Ordovician time bins thus contain a comparatively high number of short-ranging taxa (see below).

Exceptional Cambrian events are the peak origination rates at the late Terreneuvian, early Miaolingian, and early Furongian epochs. Conversely, Cambrian extinction rates peak during the middle Series 2, the late Miaolingian, and early Furongian epochs, reflecting the Botomian [40] and Marjuman extinctions [41]. During the succeeding Ordovician Period, origination rates peaked at the Dapingian–Darriwilian boundary, and extinction rates reached maximum values at the Katian–Hirnantian boundary, reflecting the LOME [42]. Lastly, Silurian extinction rates peaked at the Homerian–Gorstian boundary towards the end of that period. This reflects the Mulde event [43]. The

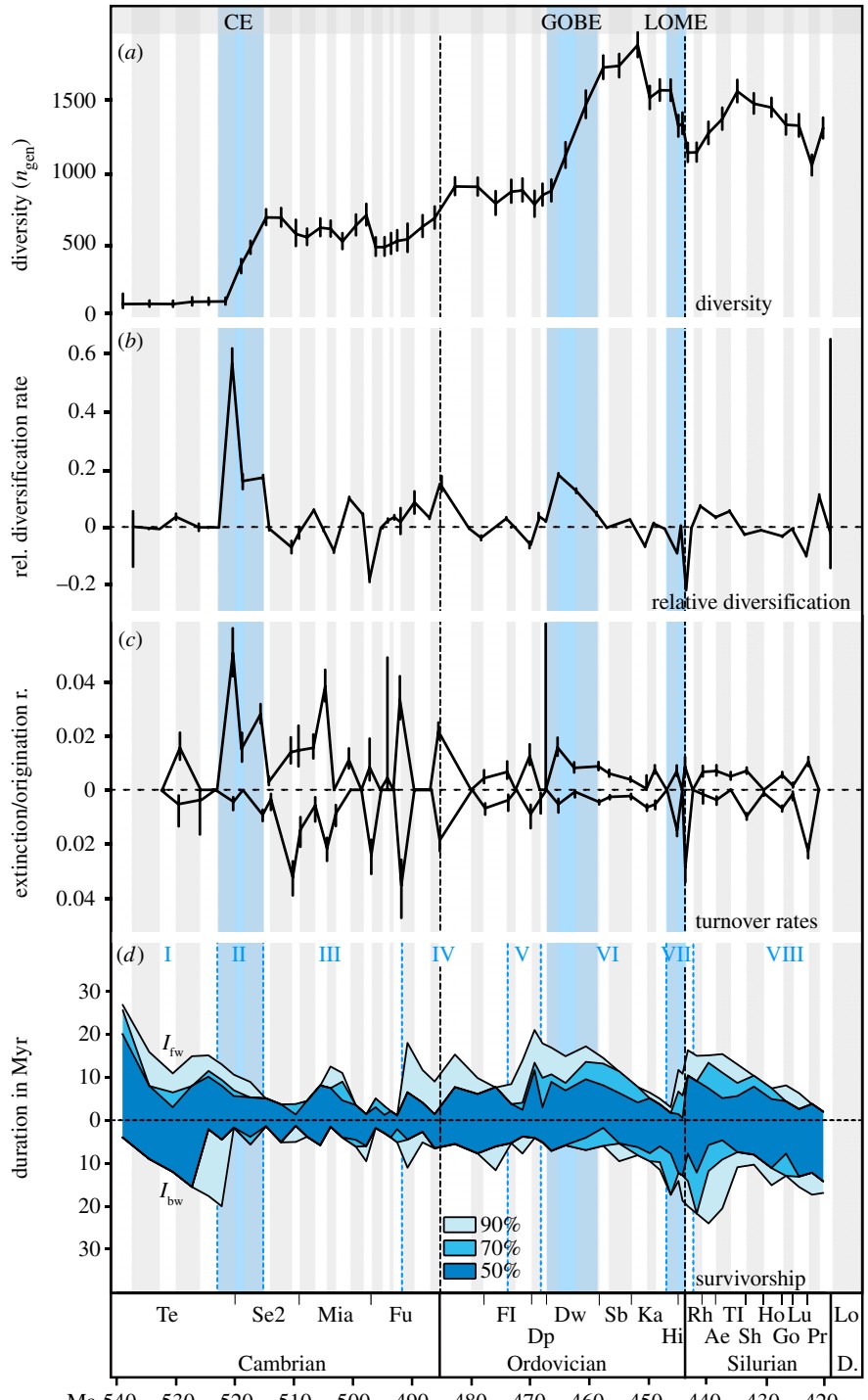

**Figure 1.** Early Palaeozoic curves of (*a*) per time bin genus level richness (adapted from [8]), (*b*) genus level relative diversification rate, (*c*) genus level extinction and origination rates (r.), and (*d*) duration of the forward ($l_{fw}$) and backward ($l_{bw}$) survivorship of 50%, 70%, and 90% of the cohort of genera of each time bin. (*a*), (*c*), and (*d*) are estimated with CR-modelling. Vertical bars indicate 95% confidence intervals. Note major changes in (*a*), (*c*), and (*d*) during the Furongian–Tremadocian interval. I-IX, designate numbered geo-historical intervals of distinct survivorship trends. Ae, Aeronian; CE, Cambrian Explosion; D., Devonian; Dp, Dapingian; Dw, Darriwilian; Fl, Floian; Fu, Furongian; Go, Gorstian; GOBE; Great Ordovician Biodiversification Event; Hi, Hirnantian; Ho, Homerian; Ka, Katian; Lo, Lochkovian; LOME; Late Ordovician Mass Extinctions; Lu, Ludfordian; Mia, Miaolingian; Pr, Pridolian; Rh, Rhuddanian; Sb, Sandbian; Se2, Cambrian Series 2; Sh, Sheinwoodian; Te, Terreneuvian; Tl, Telychian; Tr, Tremadocian.

observed events here are robust and stand out in the calculations resulting from the CR-modelling and from Alroy's [38] approach (figure 1*c*; electronic supplement material, figure S2).

## (b) Genus survivorships and longevities

The temporal variation of $l_{bw}$ and $l_{fw}$ is expressed in a geo-historical succession of eight distinct intervals (figure 1*d*), which are best described as follows: the first (I) interval is

characterized by increasing $l_{bw}$ and decreasing $l_{fw}$, reflecting the initial low diversity phase of the Terreneuvian Epoch with the appearance and slow accumulation of more and more new genera. The second (II) interval represents the CE with the rapid appearance of new genera causing $l_{bw}$ to decrease. In the third (III) interval, which lasted until the mid-Furongian Epoch, $l_{bw}$ and $l_{fw}$ remained low at values of, on average, 4–5 Myr. High taxonomic turnover during this time indicates rapid evolutionary change. In the fourth

(IV) interval, which spans the late Furongian Epoch to middle Floian Age, $l_{bw}$ and $l_{fw}$ initially increased and remained at intermediate levels. Hence, during this time more genera persisted for longer and had higher chances to survive for longer times in the future. The overall post-Terreneuvian peak of $l_{fw}$ was reached during the Dapingian Age with more than 21 Myr of 90% life durations, during the fifth (V) interval. The end of the fifth interval marks the beginning of the GOBE. Peak diversification was reached at the beginning of the sixth (VI) interval during the early Darriwilian Age and was paralleled with a decreasing $l_{fw}$. In the sixth interval, which ranges until the late Katian Age, $l_{fw}$ decreased while $l_{bw}$ increased. Hence, more and more genera occurred with long antecedent life histories, but at the same time the prospect for their future survival decreased. This is clearly an effect of the seventh (VII) interval which lasted from the latest Katian towards the early Rhuddanian Age and which represents the LOME and its direct aftermath. As a consequence of the extinctions, the age structure of the occurring genera was strongly altered and the $l_{bw}$ was at its early Palaeozoic peak in the next interval (VIII) (Rhuddanian–Sheinwoodian ages). A trend of increasing $l_{bw}$ and decreasing $l_{fw}$ during this interval indicates recovery and the appearance of more and more new genera.

## 4. Discussion

### (a) Periods of early Palaeozoic biodiversity accumulation

The synoptic comparison of evolutionary rates, survivorships, and longevity curves allows a periodization of the early Palaeozoic time into a number of intervals characterized by specific evolutionary dynamics. These intervals can be related to the known changes of the biodiversity curve and to the changes in global temperature and oxygen levels. The resulting picture of such a comparison reveals an evolutionary history that began with a relatively stable interval with low evolutionary rates, high genus survivorships, and low diversities (figure 1d, interval I). This relatively stable situation quickly escalated with the rapid appearance of skeletal lophotrochozoans, ecdysozoans, as well as sponge and archaeocyathid reefs and with a climax of the CE during the latest Terreneuvian and Cambrian Epoch 2 [44–46]. The remainder of the Cambrian was characterized by a high volatility of the evolutionary rates, extremely low genus survivorships, and a biodiversity accumulation with a rising trend towards the Ordovician (figure 1d, interval III). Our analysis thus portrays the late middle–late Cambrian as a highly dynamic period with low ecosystem resilience and this is concurrent with a growing body of evidence that the post-CE Cambrian age was a time with recurrent expansions of oxygen minimum zones across the shallow shelf and correspondent habitat disruptions [15,16,47,48].

The terminal Cambrian and the beginning of the Ordovician periods mark another phase in the evolutionary dynamics of the early Palaeozoic that lasted until the end of the Floian Age (figure 1d, intervals IV–V), which was characterized by lowered evolutionary rates, increasing genus survivorships, and a stable level in biodiversity accumulation. This interval coincides with the global expansion and biodiversification of planktic primary producers and with the first appearance of planktic graptolites and cephalopods (the 'plankton revolution' [17]).

The Middle Ordovician time records the main phase of the GOBE with a massive increase in biodiversity accumulation during the Darriwilian Age [8]. Notably, the GOBE peak diversification is preceded by a drastic increase of the forward survivorship rates of genera during the Dapingian Age. This means that Dapingian genera that survived into the Darriwilian had exceptionally high chances to further persist several Myr into the Late Ordovician. Hence, the exceptionally long-life expectancy of Dapingian genera is best explained as an *ex-post* effect of the GOBE. Similarly, the decreasing life expectancy from the Dapingian Age onward until the end of the Ordovician Period is an *ex-post* effect of the LOME. Middle and Late Ordovician genera, successively were doomed to extinction during the LOME. During the LOME genera with short precedent life histories went preferentially extinct. This is consistent with the finding that it was particularly the brachiopod genera with limited geographical ranges that went extinct [49] and that predominantly rare graptolite genera were hit already early on during the LOME [50].

At the same time, genera newly evolving and surviving during the LOME (interval VII, figure 1d) had higher chances to survive for longer. This is the well-known effect of increased life expectancies of genera occurring and originating during and immediately after mass extinctions [51,52]. Previous studies and models show that genera surviving or originating during mass extinctions tend to have a temporal advance to accumulate species [52]. As a consequence, extinctions acted as a filter for long-living genera, causing an early Palaeozoic genus longevity maximum during the Early Silurian Period. Only with the subsequent origination of new short-living Silurian genera during the post-LOME recovery, the genus longevity levels returned to pre-LOME values.

### (b) Mechanisms of early Palaeozoic biodiversity accumulation

The existence of an early Palaeozoic maximum in life expectancy ($l_{fw}$) just before the onset of GOBE is important evidence for the mechanisms behind biodiversity accumulation during this time: the GOBE coincides with an Ordovician peak in origination rates, but not with exceptionally low extinction rates (figure 1c). The exceptionally high life expectancy of Dapingian–early Darriwilian genera, therefore, cannot be explained by lowered extinction rates but as an effect of increasingly long lives of genera that did not go extinct. Dapingian–early Darriwilian genera, which persisted, did so for exceptionally long time intervals. Importantly, the Early Ordovician trend of increasing $l_{fw}$ is succeeded by a Middle–Late Ordovician trend of increasing $l_{bw}$, resulting in a massive rise in $l_o$ across the entire Ordovician. This means that, despite the environmental perturbations during the LOME, the combined life expectancy and the age structure of genera increased significantly.

The second important conclusion that can be drawn from the pre-GOBE $l_{fw}$ peak is that the maximum life expectancy was not exclusively caused by 'GOBE-specific' novel genera, but also by genera that existed well before the GOBE during the Floian and Dapingian ages. The prolonged life expectancy was not an effect of specific novel genera but an effect of an increased ability of the GOBE ecosystems to sustainably maintain existing genera.

This suggests mechanisms of ecosystem evolution during most of the Ordovician Period, where existing

genera became increasingly successfully integrated under novel ecological conditions such as different temperature and oxygenation regimes while new genera appeared constantly. One example of such an integrative mode of ecosystem evolution is the Ordovician diversification of bryozoan, coral, and stromatoporoid reefs. These clades existed as minor components in tropical shallow-water habitats from the Tremadocian Age, but collectively diversified and became dominant reef builders under cooling climatic conditions during the Middle Ordovician [12,53]. Once established, these reef builders persisted throughout the early Palaeozoic and survived even massive perturbations, such as the LOME [54].

Here, a basic difference between the Cambrian and Ordovician evolutionary dynamics becomes apparent. Cambrian conditions, such as poor oxygenation and high global temperatures are considered to be major factors of ecosystem disruptions that caused origination and extinction rates to fluctuate and genus persistence in ecosystems to decrease, e.g. [47,48]. By contrast, climatically induced global disruptions of the marine ecosystems during the LOME (e.g. [55,56]), had the opposite effect on genus persistence. During the latest Ordovician, genus longevities continued to rise even under drastically reduced biodiversity (figure 1). This basic difference is evidence of a new level of ecosystem resilience that evolved during the Ordovician. It is tempting to suggest that the Early Ordovician revolution in plankton with a first establishment of diverse and stable pelagic food chains that involved common macropredators, such as cephalopods, was an important step towards these new levels. Stable pelagic food webs affected, e.g. larval dispersal and spatial taxon ranges, which in turn potentially affected the taxon longevity.

One general conclusion can be drawn from these geo-historically more specific interpretations: generic life expectancies during the Palaeozoic were highest during time intervals directly preceding diversifications and early during diversification peaks. The diversifications affected novel and established genera likewise by increasing their average life expectancies. Therefore, processes that led to increased levels of ecosystem resilience were major factors of marine biodiversity accumulation of the Palaeozoic.

Data accessibility. The complete algorithm and relevant data are recorded in R-code and are available at https://doi.org/10.5281/zenodo.3365505.

Authors' contributions. B.K. conceived the presented idea. B.K. and F.F. performed the analysis. B.K., F.F., and C.M.Ø.R. wrote the manuscript.

Competing interests. We declare we have no competing interests

Acknowledgements. B.K. is grateful to Lee Hsiang Liow (Oslo) for encouragement to conduct CR analysis and to Susan Scholze (Helsinki) for support with data compilation with the Paleobiology Database. B.K. was funded by the Academy of Finland. C.M.Ø.R. is grateful for funding received through the VILLUM Foundation's Young Investigator Programme (grant no. VKR023452), and GeoCenter Denmark (grant nos. 2015-5 and 3-2017). F.F. is grateful to Lee Hsiang Liow, and received funding through NFR project 235073/F20 ( principal investigator: L.H.L.) and the Natural History Museum of Oslo. We further thank Amelia Penny (Helsinki) for constructive comments on an earlier version of the manuscript. Supporting data for this publication can be found in the supplementary material. This is a contribution to the IGCP Project 653 'The Onset of the Great Ordovician Biodiversification Event'.

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
