## [Reviewer comments · Proceedings of the Royal Society B: Biological Sciences]

Review History

RSPB-2019-1090.R0 (Original submission)

Review form: Reviewer 1 (Michael J. Benton)

Recommendation

Accept with minor revision (please list in comments)

Scientific importance: Is the manuscript an original and important contribution to its field?

Excellent

General interest: Is the paper of sufficient general interest?

Excellent

Quality of the paper: Is the overall quality of the paper suitable?

Excellent

Is the length of the paper justified?

Yes

Should the paper be seen by a specialist statistical reviewer?

Yes

Do you have any concerns about statistical analyses in this paper? If so, please specify them explicitly in your report.

No

It is a condition of publication that authors make their supporting data, code and materials available - either as supplementary material or hosted in an external repository. Please rate, if applicable, the supporting data on the following criteria.

Is it accessible?

Yes

Is it clear?

Yes

Is it adequate?

Yes

Do you have any ethical concerns with this paper?

Yes

Comments to the Author

The claim of this paper is an important one, that origination-extinction volatility, and with it, longevity of genera, increased markedly from Cambrian to Ordovician-Silurian. If true, this is a major discovery about a double phase of explosive diversification of marine life (Cambrian Explosion, Ordovician Biodiversification Event), moving from rapid experimentation and expansion of ecospace to stability of a sort.

The key to all this is a new time scale by Rasmussen (already published, 2019, PNAS) which corresponds to bins of 2.3 Myr duration (rather than the usual PBDB duration of 10-11 Myr). The hand work of assigning 173,293 collection records to the new time scale must have been huge, but it provides a vetted data set that is presumably more trustworthy than the raw PBDB data, and also, most importantly, precisely divided into bins. The bin length matters because many genera have durations of less than the PBDB standard bin length of 10-11 Myr.

The statistical approach using capture-recapture takes account of missing data (otherwise genus durations would be artificially curtailed at either or both ends). The modelling approach enables presentation of most likely diversity, origination and extinction, as well as generic duration, for each of the 53 time bins, and it appears to be sound, being based on widely accepted methods in ecology and paleoecology.

In Results, the high early Cambrian rates are mentioned (lines 98-101), but maybe more should be said about the fact these are partly statistical edge effects, but partly a reflection of patchy sampling, small sample size, and uncertain taxonomy of the groups sampled.

I think this is a neat paper, well written, and clear, It has involved a substantial empirical effort with the data that most analysts hitherto have been unwilling to deploy, and the statistical approaches appear to be sound and to do what is claimed.

On a small note, the authors need to pay attention to paragraphing. For example, it is not clear whether we have new paragraphs commencing at lines 25, 30, and 35, or not. The authors should

learn about the tab key on their keyboards (□) - this allows you to mark the opening of a paragraph with a standard tab inset.

- 91 Alroys = Alroy's
 93-95 Because...bins. This is not a sentence; revise.
 141 Interpretation = Discussion [?]
 146 oxygen-levels =oxygen levels
 149 lochotrochozoans = lophotrochozoans
 150 archaocyathid = archaeocyathid
 154 portraits = portrays

I note that Figures 1d and 2b are identical, and taken from Rasmussen et al. (2019), but this can be justified in that it is a reference for comparison with the other curves.

Review form: Reviewer 2

Recommendation

Major revision is needed (please make suggestions in comments)

Scientific importance: Is the manuscript an original and important contribution to its field?

Acceptable

General interest: Is the paper of sufficient general interest?

Good

Quality of the paper: Is the overall quality of the paper suitable?

Marginal

Is the length of the paper justified?

Yes

Should the paper be seen by a specialist statistical reviewer?

No

Do you have any concerns about statistical analyses in this paper? If so, please specify them explicitly in your report.

Yes

It is a condition of publication that authors make their supporting data, code and materials available - either as supplementary material or hosted in an external repository. Please rate, if applicable, the supporting data on the following criteria.

Is it accessible?

Yes

Is it clear?

Yes

Is it adequate?

Yes

Do you have any ethical concerns with this paper?

No

Comments to the Author

This paper confirms the decades-long observation that rates of origination and extinction decreased over the course of the Paleozoic (although the details of that decline have varied from study to study). A novel result, I think, is the notion that a single Cambro-Ordovician "cutpoint" can account for the transition, and that this cutpoint is found for origination rather than extinction. (For a similar result involving trilobites only, see Fig. 7 of Foote, 1988, *Paleobiology* 14:258.) The discussion of the relationship between the time series of extinction and the evolution of average longevity is also nuanced, with results I would not have predicted on the basis of prior work. (At the same time, it is absolutely necessary that average longevity must increase as extinction rate decreases, so the authors could do more to explain how the two analyses are complementary rather than redundant---it concerns not trends in averages but the effects of particular events in the time series.) It would be good for the readers of *ProcB* to see these results discussed in the context of questions of general interest. I can follow the discussion, and I care because I am interested in taxonomic rates, but the authors could do more to entrain a broader readership. The interpretations in terms of changes in ecological structure are speculative but not implausible.

The authors do not give proper consideration to previous work, when they state, "Only few studies on Palaeozoic evolutionary rates exist [4, 21–27] and only one accounts for longevity [28]." Here's a short list of papers addressing these questions---just the few that I could think of off the top of my head in a couple of minutes. No doubt some scholarly digging would reveal more.

Selected papers that analyze origination and/or extinction rates during the Paleozoic:

Raup and Sepkoski 1982, *Science*
 Raup and Boyajian 1988, *Paleobiology*
 Raup 1991
 Sepkoski 1987, *Science*
 Gilinsky and Bambach 1987, *Paleobiology*
 Gilinsky and Good 1991, *Paleobiology*
 Stanley and Powell 2003, *Geology*
 Foote 1994, *Paleobiology*
 Foote 2005, *Paleobiology*
 Foote 2007, *Paleobiology*
 Peters and Foote 2002, *Nature*
 Alroy 2008, *PNAS*
 Alroy 2010, *Palaeontology*

Selected papers that also study Paleozoic longevity per se:

Raup 1978 (cohort analysis), *Paleobiology*
 Raup 1986, *Science*
 Raup 1991, *Paleobiology*
 Baumiller 1993, *Paleobiology*
 Foote 1988, *Paleobiology*
 Foote 1991 in *Evolutionary Paleobiology* (Jablonski, Erwin, and Lipps, eds.)
 Foote and Miller 2013, *Paleobiology*

The paper also has a number of analytical issues that need to be addressed.

1. Estimates of diversity and taxonomic rates are based on capture-mark-recapture analysis, which attempts to take incomplete sampling into account. However, the description of how longevities are tabulated implies that the stratigraphic ranges are taken as good face-value estimates of true durations, with sampling not taken into account. I'm not sure how to resolve the discrepancy, because much of the discussion of longevities implies that taxa differ from one another, yet the CMR approaches, in their simplest form, assume the same transition probabilities for all taxa within a time bin. Perhaps it would give some insight if diversity, rates, and longevity were all based on the raw stratigraphic ranges? Or perhaps the CMR analysis could be modified to allow rates to be taxon-specific and/or depend on genus age (i.e. time since genus origin)?

2. The decline in forward longevity through the Silurian must be an artifact. The description of the data download says that the time range of occurrences is limited to the Cambrian through Lochkovian. So genera may in fact persist past the Lochkovian, but their durations would be truncated. (The same holds for backward survivorship in the Cambrian.) This is not a serious issue, but it should be acknowledged.

3. I cannot reconcile the transition probabilities in Fig. 1(b) with the rates in Fig. S2(b). If P is the probability and T is the interval length, the rate is simply equal to $-\log(1-P)/T$ (suppl. ref. 6). Consider the highest peak in the origination in the Cambrian, with a probability of about 0.45. The interval (Te_5) is 2.1 Myr long according to ref. 9. So the rate should be equal to about 0.28. Yet Fig. S2(b) shows it to be closer to 0.05. The same discrepancy holds for other intervals.

Decision letter (RSPB-2019-1090.R0)

10-Jun-2019

Dear Dr Kröger:

I am writing to inform you that your manuscript RSPB-2019-1090 entitled "The evolutionary dynamics of the early Palaeozoic marine biodiversity accumulation" has, in its current form, been rejected for publication in Proceedings B.

This action has been taken on the advice of referees, who have recommended that substantial revisions are necessary. With this in mind we would be happy to consider a resubmission, provided the comments of the referees are fully addressed. However please note that this is not a provisional acceptance.

1) A 'response to referees' document including details of how you have responded to the comments, and the adjustments you have made.

- 2) A clean copy of the manuscript and one with 'tracked changes' indicating your 'response to referees' comments document.
- 3) Line numbers in your main document.

Sincerely,

Dr Daniel Costa
 mailto: proceedingsb@royalsociety.org

Associate Editor
 Board Member: 1
 Comments to Author:

Both reviewers provide constructive critiques of the ms, with Rev1 more positive than Rev2. Rev2's main concerns have to do with scholarship and analysis. The authors simply do not cite the relevant prior work. Moreover, several analyses need to be re-visited. We will be in a better position to judge the outcome for this ms if the authors can satisfy the concerns of Rev 2.

Reviewer(s)' Comments to Author:

Referee: 1

Comments to the Author(s)

The claim of this paper is an important one, that origination-extinction volatility, and with it, longevity of genera, increased markedly from Cambrian to Ordovician-Silurian. If true, this is a major discovery about a double phase of explosive diversification of marine life (Cambrian Explosion, Ordovician Biodiversification Event), moving from rapid experimentation and expansion of ecospace to stability of a sort.

The key to all this is a new time scale by Rasmussen (already published, 2019, PNAS) which corresponds to bins of 2.3 Myr duration (rather than the usual PBDB duration of 10-11 Myr). The hand work of assigning 173,293 collection records to the new time scale must have been huge, but it provides a vetted data set that is presumably more trustworthy than the raw PBDB data, and also, most importantly, precisely divided into bins. The bin length matters because many genera have durations of less than the PBDB standard bin length of 10-11 Myr.

The statistical approach using capture-recapture takes account of missing data (otherwise genus durations would be artificially curtailed at either or both ends). The modelling approach enables presentation of most likely diversity, origination and extinction, as well as generic duration, for each of the 53 time bins, and it appears to be sound, being based on widely accepted methods in ecology and paleoecology.

In Results, the high early Cambrian rates are mentioned (lines 98-101), but maybe more should be said about the fact these are partly statistical edge effects, but partly a reflection of patchy sampling, small sample size, and uncertain taxonomy of the groups sampled.

I think this is a neat paper, well written, and clear, It has involved a substantial empirical effort

with the data that most analysts hitherto have been unwilling to deploy, and the statistical approaches appear to be sound and to do what is claimed.

On a small note, the authors need to pay attention to paragraphing. For example, it is not clear whether we have new paragraphs commencing at lines 25, 30, and 35, or not. The authors should learn about the tab key on their keyboards (□) - this allows you to mark the opening of a paragraph with a standard tab inset.

- 91 Alroys = Alroy's
- 93-95 Because...bins. This is not a sentence; revise.
- 141 Interpretation = Discussion [?]
- 146 oxygen-levels =oxygen levels
- 149 lochotrochozoans = lophotrochozoans
- 150 archaocyathid = archaeocyathid
- 154 portraits = portrays

I note that Figures 1d and 2b are identical, and taken from Rasmussen et al. (2019), but this can be justified in that it is a reference for comparison with the other curves.

Referee: 2

Comments to the Author(s)

This paper confirms the decades-long observation that rates of origination and extinction decreased over the course of the Paleozoic (although the details of that decline have varied from study to study). A novel result, I think, is the notion that a single Cambro-Ordovician "cutpoint" can account for the transition, and that this cutpoint is found for origination rather than extinction. (For a similar result involving trilobites only, see Fig. 7 of Foote, 1988, *Paleobiology* 14:258.) The discussion of the relationship between the time series of extinction and the evolution of average longevity is also nuanced, with results I would not have predicted on the basis of prior work. (At the same time, it is absolutely necessary that average longevity must increase as extinction rate decreases, so the authors could do more to explain how the two analyses are complementary rather than redundant---it concerns not trends in averages but the effects of particular events in the tie series.) It would be good for the readers of *ProcB* to see these results discussed in the context of questions of general interest. I can follow the discussion, and I care because I am interested in taxonomic rates, but the authors could do more to entrain a broader readership. The interpretations in terms of changes in ecological structure are speculative but not implausible.

The authors do not give proper consideration to previous work, when they state, "Only few studies on Palaeozoic evolutionary rates exist [4, 21–27] and only one accounts for longevity [28]." Here's a short list of papers addressing these questions---just the few that I could think of off the top of my head in a couple of minutes. No doubt some scholarly digging would reveal more.

Selected papers that analyze origination and/or extinction rates during the Paleozoic:

- Raup and Sepkoski 1982, *Science*
- Raup and Boyajian 1988, *Paleobiology*
- Raup 1991
- Sepkoski 1987, *Science*
- Gilinsky and Bambach 1987, *Paleobiology*
- Gilinsky and Good 1991, *Paleobiology*

Stanley and Powell 2003, *Geology*
 Foote 1994, *Paleobiology*
 Foote 2005, *Paleobiology*
 Foote 2007, *Paleobiology*
 Peters and Foote 2002, *Nature*
 Alroy 2008, *PNAS*
 Alroy 2010, *Palaeontology*

Selected papers that also study Paleozoic longevity per se:

Raup 1978 (cohort analysis), *Paleobiology*
 Raup 1986, *Science*
 Raup 1991, *Paleobiology*
 Baumiller 1993, *Paleobiology*
 Foote 1988, *Paleobiology*
 Foote 1991 in *Evolutionary Paleobiology* (Jablonski, Erwin, and Lipps, eds.)
 Foote and Miller 2013, *Paleobiology*

The paper also has a number of analytical issues that need to be addressed.

1. Estimates of diversity and taxonomic rates are based on capture-mark-recapture analysis, which attempts to take incomplete sampling into account. However, the description of how longevities are tabulated implies that the stratigraphic ranges are taken as good face-value estimates of true durations, with sampling not taken into account. I'm not sure how to resolve the discrepancy, because much of the discussion of longevities implies that taxa differ from one another, yet the CMR approaches, in their simplest form, assume the same transition probabilities for all taxa within a time bin. Perhaps it would give some insight if diversity, rates, and longevity were all based on the raw stratigraphic ranges? Or perhaps the CMR analysis could be modified to allow rates to be taxon-specific and/or depend on genus age (i.e. time since genus origin)?
2. The decline in forward longevity through the Silurian must be an artifact. The description of the data download says that the time range of occurrences is limited to the Cambrian through Lochkovian. So genera may in fact persist past the Lochkovian, but their durations would be truncated. (The same holds for backward survivorship in the Cambrian.) This is not a serious issue, but it should be acknowledged.
3. I cannot reconcile the transition probabilities in Fig. 1(b) with the rates in Fig. S2(b). If P is the probability and T is the interval length, the rate is simply equal to $-\log(1-P)/T$ (suppl. ref. 6). Consider the highest peak in the origination in the Cambrian, with a probability of about 0.45. The interval (Te_5) is 2.1 Myr long according to ref. 9. So the rate should be equal to about 0.28. Yet Fig. S2(b) shows it to be closer to 0.05. The same discrepancy holds for other intervals.

Author's Response to Decision Letter for (RSPB-2019-1090.R0)

See Appendix A.

RSPB-2019-1634.R0

Review form: Reviewer 2

Recommendation

Accept with minor revision (please list in comments)

Scientific importance: Is the manuscript an original and important contribution to its field?

Good

General interest: Is the paper of sufficient general interest?

Good

Quality of the paper: Is the overall quality of the paper suitable?

Good

Is the length of the paper justified?

Yes

Should the paper be seen by a specialist statistical reviewer?

No

Do you have any concerns about statistical analyses in this paper? If so, please specify them explicitly in your report.

No

It is a condition of publication that authors make their supporting data, code and materials available - either as supplementary material or hosted in an external repository. Please rate, if applicable, the supporting data on the following criteria.

Is it accessible?

Yes

Is it clear?

Yes

Is it adequate?

Yes

Do you have any ethical concerns with this paper?

No

Comments to the Author

I believe the authors have effectively addressed concerns I raised in my initial review.

If I understand correctly, the PBDB-DL#2 contains occurrence data extending well after the Silurian. If that is the case, then lower survivorship of later Silurian cohorts is not an edge effect.

The apparent discrepancy between rates and probabilities has been explained, but there is still an inconsistency between Fig. 1(c) and Fig. S2(b). The curves in the two figures are the same, as the authors point out, but the values shown on the ordinate differ by an order of magnitude (0.2 vs. 0.02, etc.).

I believe the URL for DL#2 (line 14 of the supplementary text) needs to be corrected. When I type that URL in to try to re-create the download, I get the following message:

400 Bad Request

- you must specify at least one of the parameters 'all_records', 'occ_id', 'coll_id', 'clust_id', 'coll_match', 'coll_re', 'base_name', 'taxon_name', 'match_name', 'base_id', 'taxon_id', 'idreso', 'abundance', 'lngmin', 'lngmax', 'latmin', 'latmax', 'loc', 'plate', 'cc', 'state', 'county', 'continent', 'strat', 'formation', 'stratgroup', 'member', 'lithology', 'envtype', 'interval_id', 'interval', 'min_ma', 'max_ma'

The paper will require copy-editing to clean up the English.

Decision letter (RSPB-2019-1634.R0)

20-Jul-2019

Dear Dr Kröger:

Your manuscript has now been peer reviewed and the reviews have been assessed by an Associate Editor. The reviewers' comments (not including confidential comments to the Editor) and the comments from the Associate Editor are included at the end of this email for your reference. As you will see, the reviewers and the Editors have raised some concerns with your manuscript and we would like to invite you to revise your manuscript to address them.

Research ethics:

Use of animals and field studies:

Please submit a copy of your revised paper within three weeks. If we do not hear from you within this time your manuscript will be rejected. If you are unable to meet this deadline please let us know as soon as possible, as we may be able to grant a short extension.

Best wishes,

Dr Daniel Costa
 mailto: proceedingsb@royalsociety.org

Associate Editor Board Member

Comments to Author:

Your paper has been favourable reviewed by Rev 2. However, there are a few additional comments that need to be addressed before we can accept the paper. Please address these comments in your revision.

Reviewer(s)' Comments to Author:

Referee: 2

Comments to the Author(s).

I believe the authors have effectively addressed concerns I raised in my initial review.

If I understand correctly, the PBDB-DL#2 contains occurrence data extending well after the Silurian. If that is the case, then lower survivorship of later Silurian cohorts is not an edge effect.

The apparent discrepancy between rates and probabilities has been explained, but there is still an inconsistency between Fig. 1(c) and Fig. S2(b). The curves in the two figures are the same, as the authors point out, but the values shown on the ordinate differ by an order of magnitude (0.2 vs. 0.02, etc.).

I believe the URL for DL#2 (line 14 of the supplementary text) needs to be corrected. When I type that URL in to try to re-create the download, I get the following message:

400 Bad Request

- you must specify at least one of the parameters 'all_records', 'occ_id', 'coll_id', 'clust_id', 'coll_match', 'coll_re', 'base_name', 'taxon_name', 'match_name', 'base_id', 'taxon_id', 'idreso', 'abundance', 'lngmin', 'lngmax', 'latmin', 'latmax', 'loc', 'plate', 'cc', 'state', 'county', 'continent', 'strat', 'formation', 'stratgroup', 'member', 'lithology', 'envtype', 'interval_id', 'interval', 'min_ma', 'max_ma'

The paper will require copy-editing to clean up the English.

Author's Response to Decision Letter for (RSPB-2019-1634.R0)

See Appendix B.

Decision letter (RSPB-2019-1634.R1)

06-Aug-2019

Dear Dr Kröger

I am pleased to inform you that your manuscript entitled "The evolutionary dynamics of the early Palaeozoic marine biodiversity accumulation" has been accepted for publication in Proceedings B.

Open Access

Paper charges

Sincerely,

Dr Daniel Costa

Associate Editor:

Board Member

Comments to Author:

(There are no comments.)

Appendix A

Dear Editor Dr. Daniel Costa,

Thank you very much for your decision letter sent on 10-Jun-2019 regarding our manuscript ID RSPB-2019-1090 entitled “The evolutionary dynamics of the early Palaeozoic marine biodiversity accumulation.

Here we resubmit our manuscript after substantial revision. The critiques of the reviewers and the comments of editors helped substantially to improve our manuscript.

Particularly we took focus on an improved methodology of the survivorship analysis and, as suggested by Reviewer 2, before our revision, took an in-depth review on previous work on the topic. We have therefore changed the manuscript accordingly. This resulted in a modified Introduction section (lines 30–53) and made necessary some substantial changes in the Methods section (lines 77–93).

With the improved methodology our previous results could be fully supported and essentially no changes in the results were necessary. However, based on suggestions of reviewer 2, we now compare our new results with previous analysis (Results-section lines 102–109). Our results are consistent with these earlier analyses but have a higher stratigraphic resolution and/or cover a wider range of taxa.

We changed our Discussion section only slightly, mainly taking into consideration the suggestions of reviewer 2 to better connect to questions of general interest (lines 236–241).

Please find below our detailed response to the referees.

Sincerely,
Björn Kröger

Response to referees:

Referee 1:

We followed all smaller suggestions and corrected our manuscript accordingly.

Referee 1 remarked that the diversity curve (now figure 1a) is identical to Rasmussen et al. (2019). Actually, it is not fully identical as there are very minor differences. This is, because each run of the CR-modelling is slightly different (within levels of confidence). We now discuss this explicitly in the supplementary information and add “(adapted from [8])” into the capture of figure 1.

Referee 1 remarked on the statistical edge effects. See our comment below.

Referee 2:

Referee 2 suggested we should consider to discuss Foote (1988). We do this now in the results section lines 103–109.

Referee 2 suggested to do more to explain how the evolutionary rates, survivorship analyses are complementary rather than redundant. We do this now in the introductory section in lines 36-46.

Referee 2 remarked that we gave not proper consideration to previous work in our introductory statement. After carefully reading the literature list provided by Referee 2 we abandoned our originally oversimplified statement and replaced it with a more appropriate one in the Introduction at lines 33–36 and with a more balanced selection of references.

Referee 2 additionally listed three analytical issues. This is our response:

1. Referee 2 expressed concerns with our method calculating genus longevities based on FAD's and LAD's taken as face values from the PBDB compilations. We agree with this and we found a solution to circumvent the problem. Now we do not calculate individual genus level durations. Instead, we use survivorship curves of genus cohorts of the time bins as proxies for longevities. This allows us to use capture-recapture modelling to estimate richness curves for each cohort and based on this to calculate 50%, 70%, 90% forward and backward survivorships. In this way sampling and preservation are explicitly taken into consideration and consistency with the evolutionary rates calculations are reached. The method is explained in the Methods-section (lines 88–93) and in more detail in the supplementary information.
2. Referee 2 remarked that the decline in forward longevity through the Silurian and the backward survivorship through the Cambrian must be an artefact and a statistical edge effect (see also remark from Reviewer 1). Although an edge effect, the Cambrian pattern reflects the real successive appearance and persistence of new metazoan genera. The Cambrian pattern exists not because the backward survivorship is artificially truncated at the base Cambrian, but because there were so few metazoans in the earliest Cambrian. In this sense it is part of the story, and we believe our discussion in lines 131–133 is sufficient. The Silurian pattern is more problematic to explain. In our originally submitted version our methods were not fully clear. We now improved the description of our methods (line 56–62) and detailed it in the supplementary information. Additionally, we added a paragraph, discussing the peculiar Silurian trajectory in the supplementary information. We believe the Silurian pattern is not an artefact; it may require additional scrutiny in a separate paper including the Devonian.
3. Referee 2 noted that she/he cannot reconcile transition probabilities in figure 1b with the rates in figure S2b. Actually, this problem was simply based on a typo in the original figure caption of figure 1 where we mistakenly wrote “probabilities” instead of “rates”. Both figures illustrate absolutely identical values (the rates of origination and extinction). Because we reorganised figure 1, the rates are now in figure 1c and figure S2b and the typo is corrected.

Appendix B

Dear Dr. Daniel Costa,

Thank you for your decision letter from 20-Jul-2019 and for forwarding us the comments of the second review of our submission.

We apologize for the small mistakes left in the previous version of our submission, remarked by referee 2.

Referee 2 addressed three concerns:

1. Inconsistencies between values shown in figures 1c and S2b.

We now corrected the wrong labelling of Figure 1c. The wrong numbers resulted from an oversight during redrawing of the graphic. The correct values of the evolutionary rates are in the 0.00–0.05 range.

2. Incorrect URL in line 14 of the supplementary text

We now corrected the URL and tested the download. The correct URL needs to have the stratigraphic interval specified, which is the case now.

3. Copy editing of the English

We contacted the editorial team of Proceedings B on 01-Aug-2019 and Shalene Singh-Shepherd wrote per email that at this stage no copy-editing is necessary.

Additionally, we modified the layout of the supplementary material document slightly (deleted the line numbers) and added information about the main manuscript (future DOI and journal information) below the title.

We hope that with these changes our submission can be accepted for publication in Proceedings B.

In case the manuscript will be accepted we will upload the R-code and the data to zenodo.org. This will create a DOI to this site.

With kind regards,
Björn Kröger